# Small Polar Molecules: A Challenge in Marine Chemical Ecology

**DOI:** 10.3390/molecules24010135

**Published:** 2018-12-31

**Authors:** Eva Ternon, Yanfei Wang, Kathryn J. Coyne

**Affiliations:** 1College of Earth, Ocean, and Environment, University of Delaware, 700 Pilottown Road, Lewes, DE 19958, USA; yfwang@udel.edu; 2Université Côte d’Azur, CNRS, OCA, IRD, Géoazur, 250 rue Albert Einstein, 06560 Valbonne, France

**Keywords:** allelochemicals, extraction, characterization, algicide, polyamines

## Abstract

Due to increasing evidence of key chemically mediated interactions in marine ecosystems, a real interest in the characterization of the metabolites involved in such intra and interspecific interactions has emerged over the past decade. Nevertheless, only a small number of studies have succeeded in identifying the chemical structure of compounds of interest. One reason for this low success rate is the small size and extremely polar features of many of these chemical compounds. Indeed, a major challenge in the search for active metabolites is the extraction of small polar compounds from seawater. Yet, a full characterization of those metabolites is necessary to understand the interactions they mediate. In this context, the study presented here aims to provide a methodology for the characterization of highly polar, low molecular weight compounds in a seawater matrix that could provide guidance for marine ecologists in their efforts to identify active metabolites. This methodology was applied to the investigation of the chemical structure of an algicidal compound secreted by the bacteria *Shewanella* sp. IRI-160 that was previously shown to induce programmed cell death in dinoflagellates. The results suggest that the algicidal effects may be attributed to synergistic effects of small amines (ammonium, 4-aminobutanal) derived from the catabolization of putrescine produced in large quantities (0.05–6.5 fmol/cell) by *Shewanella* sp. IRI-160.

## 1. Introduction

Marine chemical ecology emerged in the early 1980s, conducted by marine ecologists who identified interactions mediated by chemicals produced by various organisms, from small planktonic species to large invertebrates [1]. Yet, very few scientists were successful at fully characterizing metabolites responsible for the observed chemical interactions. Despite the recent collaboration between marine ecologists and natural product chemists, the identification of the structure or at least the compound class of many metabolites remains scarce [2,3]. To fully understand the ecological role and mode of action of those metabolites, they must be tested as pure compounds and therefore need to be extracted, fractionated and fully characterized.

Allelopathic interactions are now recognized to contribute to the paradoxical coexistence of a large diversity of species in an ecosystem with limited resources, evoked by [4] as the “paradox of the plankton”. In the microbial ecosystem, bacteria and microalgae secrete secondary metabolites to mediate various interactions, like competition for resources, mate attraction, and predation (reviewed by [3]). Microbial allelochemicals are diverse in structure and mode of action, and cover a wide range of polarity, from polar amino-acids [5] to apolar fatty acids or sterols [6,7,8]. They also span a wide range of size, comprising small compounds with molecular weights below 200 Da [5], intermediate molecules like toxins (reviewed in [9]), and larger functional enzymes [10]. Those compounds can either be synthesized de novo in the cells and subsequently released in the seawater, or result from the transformation of metabolites via enzymatic reactions or spontaneous degradation when released in seawater. Therefore, in some cases, the only way to access the metabolite of interest is its extraction from the seawater matrix. Extraction and further characterization of active metabolites from seawater can be extremely challenging, considering (i) the highly ionized matrix inherent to the chemistry of seawater, and (ii) the low yield of metabolites isolated that would be available for spectral analysis (e.g., Nuclear Magnetic Resonance).

Metabolite characteristics lead to a wide range of specific behavior in seawater that prevents the establishment of a universal extraction procedure. Although significant advances in the characterization of marine chemical cues from bacteria [11], microalgae [12], copepods [7], algae [12] and invertebrates [13] have been accomplished in recent decades, very few of these metabolites are of very low molecular weight [3,11] (C1–C6, <125 Da) and extreme polar features [5,14]. This extreme polarity contributes to the inability to perform chemical characterization of active metabolites in several cases [15,16,17]. Yet, several of these Small Polar Compounds (SPC) secreted by microbial organisms have been shown to play key roles in biotic interactions, such as small oxylipins [6], amino-acids (e.g., *N*-β-methylamino-l-alanine (BMMA) [18]), pheromones (e.g., l-diproline [5]), polyamines (e.g., spermine and putrescine [19]), and volatile compounds (e.g., dimethylsulfide [20]). Clearly, there is a need to resolve the analytical bottlenecks preventing the characterization of SPC (<125 Da) dissolved in seawater.

In this context, classic methods for extraction (i.e., liquid-liquid extraction (LLE) and solid phase extraction (SPE)) of these metabolites have little value, and specific analytical methods must be developed to target the metabolite of interest. Comparative mass spectrometry-based (MS) metabolomics was recently presented as a good substitute for the characterization of metabolites, since it does not require fractionation or isolation steps [5,21]; however, extraction from seawater matrices makes the use of fractionation tools compulsory in this case.

A decade ago, Hare et al. [22] observed inhibiting effects induced by the presence of a bacterium, *Shewanella* sp. IRI-160, on the growth of several species of dinoflagellates (*Pfiesteria piscicida*, *Prorocentrum minimum*, *Gyrodinium uncatenum*). Pokrzywinski [16] found that the algicide produced by *Shewanella* was released into the growth medium. Exposure to this algicide, called IRI-160AA, induced a biochemical response consistent with programmed cell death (PCD) in several species of dinoflagellates [23], accompanied by significant morphological changes [24] and impacts on dinoflagellate photo-physiology [25]. In terms of basic features, IRI-160AA was listed as a “small, hydrophilic, and thermally stable compound” (i.e., SPC [16]), while further characterization of its structure remained unsuccessful. Additional information on IRI-160AA was recently acquired, providing new insights on the structure of this metabolite. This paper presents those new data, together with a step-by-step analytical approach that can be followed in the context of the search of bioactive SPC. Indeed, to our knowledge, only a couple of studies have succeeded with the characterization of bioactive SPC [5,14], suggesting that the approach presented here would be of value to marine ecologists when facing the characterization of SPC.

## 2. Results

The five-step analytical procedure used to characterize the algicidal compound released by the bacteria *Shewanella* sp. IRI160 is described here.

Step 1—Since each analytical step (extraction, fractionation) has to be validated before proceeding to the next one, the implementation of a simple, quick and reliable bioassay is essential. A bioassay testing the activity of IRI-160AA was developed by Pokrzywinski et al. [16] and consisted of adding the filtered bacterial medium to the culture of dinoflagellates in a 4–10% (*v*/*v*) proportion. *Rhodomonas* sp. was used as a control negative-response species. The physiological status of the dinoflagellates and *Rhodomonas* sp. were assessed after 24 h by measuring the in vivo chlorophyll fluorescence. This measurement only takes a couple of minutes per sample and is highly reliable. Therefore, as an excellent example of a simple, quickly implemented and reliable bioassay, it was used in our efforts to isolate compound(s) responsible for the algicide activity.

Step 2—Several structural features, including size, resistance to heat and light, and stability over time, can be investigated to determine the “face of a molecule” [26]. Thus, it is of importance to first determine physico-chemical characteristics that may guide the extraction. IRI-160AA was shown to be stable over a large range of temperatures and in particular when autoclaved 20 min at 120 °C or boiled for 20 min in a closed container [16]. The resistance to high temperatures rules out several families of metabolites that are sensitive to heat, such as proteins, peptides, and several vitamins. The findings reported here, however, indicated a complete loss of activity for IRI-160AA when the bacterial exudate was boiled or autoclaved in an open container (Figure 1a). This result suggests that although the algicide is stable at high temperature, it is prone to volatilization. Further use of dialysis (100–500 Da cut-off) also resulted in a loss of activity in IRI-160AA, confirming that the metabolite(s) of interest is a low molecular weight compound (<100 Da, Figure 1b). Additionally, it is worth noting that organic compounds may release specific odors according to their chemical family; some alcohols recall freshly cut grass, alkanes may smell like gasoline, and amines can have a strong fish odor. Here, the algicide presented an odor similar to decaying fish. These characteristics together suggested that the algicide included a low molecular weight volatile amine, and the extraction and further characterization was further directed towards this hypothesis.

Step 3—Organic compounds can be extracted from seawater using classic LLE or SPE techniques, although the latter is usually preferred when working with large volumes of water. SPE may be performed using either cartridges or disks that are available with a wide variety of phases (e.g., silica C18 and C8, ionic exchange, divinyl styrene benzene). The choice of phase is motivated by the nature of the compound and its resulting interactions with the phase. In the search of unknown compounds, starting with silica C18 will cover a larger diversity of metabolites. Preliminary extractions of IRI-160AA algicide performed using silica C18 however resulted in 100% recovery of algicidal activity within the pass-through, while any non-polar constituents were likely retained by the C18 [16]. Therefore, alternative phases including ion exchange, activated carbon or hydrophilic-lipophilic balanced cartridges had to be considered. The high amount of salts in seawater can make the use of ionic exchange difficult; hence, hydrophilic-lipophilic balanced cartridges were used, first at unmodified pH and subsequently at pH 3 and 11. The algicide that had passed through the cartridge was subsequently neutralized to pH 7.5 and further tested for its activity using the bioassay. Loss of activity was observed for both pH 3 and 11, but was only significantly so at pH 11 (*p* < 0.01; Figure 1c). Basic pH would charge acids and leave amines under their neutral forms, which would then be retained on the cartridge. This result supports the hypothesis of a bioactive metabolite belonging to the amines family of compounds. Yet, its polarity and small size still precluded an efficient extraction using classic SPE, and complementary techniques had to be used.

Step 4—The extraction of metabolites smaller than 100 Da on SPE cartridges can be facilitated by use of derivative agents [27,28,29,30]. Through simple chemical reaction, the derivative agent and the metabolite of interest form a larger molecule easily extractable and detectable by either UV or fluorescence detectors. The derivative agent must be selected based on the main functional characteristics of the metabolite of interest (e.g., carbonyl, amine, etc.), and the technique of separation (Gas Chromatography (GC) or Liquid Chromatography (LC)). Since preliminary experiments suggested affiliations of IRI-160AA to the amines, dansyl chloride (**1**) was chosen as the derivative agent (Figure 2). Analysis of both derivatized blank (seawater) and algicide samples by HPLC-fluorescence showed two shared peaks as well as a few additional peaks specific to IRI-160AA (Figure 3a).

Shared peaks were identified as dansylated hydroxyl and dansylated glycine resulting from the reaction with free-OH and the glycine that was added in the samples. Extra peaks were identified as dansylated ammonia (**2**) and *n*-butylamine after confirmation with standards. This was further confirmed by Ultra High-Performance Liquid Chromatography (UHPLC) coupled to a High-Resolution Mass Spectrometry (HRMS), undertaking fragmentation (MSMS) experiments (Figure 3b and c; *m*/*z* 251.08499, and *m*/*z* 307.14749, respectively). Analysis of the derivatized cellular extract by UHPLC-HRMS-MSMS showed the presence of a major peak (Figure 4a) further identified as dansylated putrescine (*m*/*z* 303.11484, Figure 4b). Dansylated ammonia was present at very low intensities in cell extracts, suggesting this metabolite was excreted and not stored in cells. Dansylated 4-aminobutanal was also identified in the cell extract, producing the accurate *m*/*z* 361.15689 from full chromatogram (Figure 3c).

Step 5—To confirm the relevance of the chemical findings, the activity of standards of pure *n*-butylamine and a mixture of ammonia and *n*-butylamine, both detected in the IRI-160AA, were tested on the growth of two dinoflagellates *Prorocentrum minimum* and *Gyrodinium instriatum*, as well as *Rhodomonas* sp., using the following concentrations: 20 and 200 µM of *n*-butylamine both with and without addition of 100 µM of ammonia. *Rhodomonas* sp. constitutes the control species that in previous assays showed no sensitivity to IRI-160AA [16]. It is worth noting that preliminary investigations showed that the growth of *G. instriatum* significantly decreased with addition of 100 µM of ammonium alone (*p* < 0.05), where the relative fluorescence of *G. instriatum* exposed to ammonium at this concentration was about 0.8 of controls (not shown). There was no significant effect of ammonium on the growth of *P. minimum* or *Rhodomonas* sp. at this concentration.

The addition of *n*-butylamine alone had significantly negative impacts on all species tested and was dose-dependent (*p* < 0.05) (Figure 5). Relative fluorescence of *P. minimum*, *G. instriatum*, and *Rhodomonas* sp. was 5.16, 3.16, and 1.32 times higher in the treatments with 20 µM *n*-butylamine compared to the same species with 200 µM *n*-butylamine, respectively. Preliminary data indicates that *Rhodomonas* sp. exhibited a significantly greater sensitivity than dinoflagellates to addition of 20 µM *n*-butylamine (*p* < 0.05), while addition of *n*-butylamine at concentrations equal to or higher than 200 µM had a significantly greater effect on dinoflagellate cell density than on *Rhodomonas* sp. (*p* < 0.05) (Figure 1, Appendix A). Here, the results were consistent with these previous experiments, showing that dinoflagellate growth was more impaired by higher concentrations of *n*-butylamine than the control species. Indeed, the addition of 200 µM of *n*-butylamine yielded a significantly lower fluorescence of both *P. minimum* and *G. instriatum* than *Rhodomonas* sp. relative to controls (2.68 and 2.08 times lower respectively, *p* < 0.05), whereas at 20 µM of *n*-butylamine, the relative fluorescence of the dinoflagellates was higher than *Rhodomonas* sp. (1.46 and 1.14 respectively, *p* < 0.05 and *p* > 0.05 respectively). Dinoflagellate species had a similar response to a 20 µM addition of *n*-butylamine, showing no significant difference in relative fluorescence, while *G. instriatum* was more impacted by the addition of 200 µM of *n*-butylamine than *P. minimum*, with a relative fluorescence 1.29 significantly higher (*p* < 0.05).

Synergistic effects of ammonium and *n*-butylamine were further investigated by adding 100 µM of ammonium to the 20 and 200 µM *n*-butylamine solutions. Interestingly, the addition of both ammonium and *n*-butylamine resulted in a significant decrease of the relative fluorescence of the dinoflagellates (*p* < 0.05, Figure 5), at both 20 and 200 µM *n*-butylamine concentrations. The decrease of the fluorescence of *P. minimum* and *G. instriatum* relative to controls was more marked at 20 µM of *n*-butylamine (1.45 and 1.85 times, respectively) than at 200 µM (1.37 and 1.47, respectively), indicating that at high concentrations, the effects of *n*-butylamine prevail on the effects of ammonium. The opposite pattern was observed for *Rhodomonas* sp. since the addition of ammonium yielded a significant increase in relative fluorescence at 200 µM of *n*-butylamine (by a factor 1.23, *p* < 0.05), and a noticeable, although non-significant, increase in relative fluorescence at 20 µM of *n*-butylamine. This result indicates that an addition of ammonium at 100 µM may benefit this species.

*G. instriatum* was the most sensitive species to the synergistic action of ammonium and low concentrations of *n*-butylamine (20 µM), showing a significantly lower relative fluorescence by 1.75 and 1.63 than *Rhodomonas* sp. and *P. minimum*, respectively. At high concentrations of *n*-butylamine (200 µM), both dinoflagellates were equally affected by the synergistic action of the two substances (*p* > 0.05) and were shown to be much more sensitive than *Rhodomonas* sp., which showed a significantly higher relative fluorescence, by 4.52 and 3.78, respectively, than *P. minimum* and *G. instriatum*.

## 4. Discussion

The bacteria *Shewanella* sp. IRI-160 releases one or several compound(s) into the culture medium that impairs the growth of several dinoflagellates [16,25]. The metabolite(s) of interest being SPC [16], their extraction from the seawater matrix using classic techniques of SPE has remained unsuccessful, making cutting-edge tools like metabolomics challenging. The acquisition of several physico-chemical characteristics (pH, volatility and size), however, guided our search towards the low molecular weight amines, and dansyl chloride was further used to derivatize the compounds for detection by HPLC-fluorescence and UHPLC-HRMS.

The derivatization of the cells revealed the production of significant amounts of the polyamine putrescine (Figure 2, (**4**), 0.05–6.5 fmol/cell), a common feature of *Shewanella* species (0.01–3 µmol/g wet cells in [31]). On the other hand, the derivatized cell-free medium contains substantial amounts of ammonium (0.77–7.26 mM) and *n*-butylamine ((**3**), 2.50–8.89 µM), two SPC amines, while putrescine was not detected. Polyamines are suspected to have a role in early and/or last stages of programmed cell death (PCD) in plants, among other important physiological functions [32]. PCD can be induced by either a direct action of the polyamines, through their regulatory effect on ion channels (K^+^, Ca^2+^), or by an indirect action of their catabolism which produces intermediates such as reactive oxygen species (ROS) like H_2_O_2_, and aminoaldehydes [33]. Since recent work by Pokrwzynsky [23] provides evidence that exposure to IRI-160AA induces PCD to several dinoflagellates, indirect effects of polyamines may be suspected. In plants, the oxidation of putrescine mediated by diamine oxidase yields 4-aminobutanal, H_2_O_2_, and ammonium (NH_4_^+^) [34], supporting the hypothesis of a catabolization of putrescine upon release into the medium by *Shewanella* sp. IRI-160. Following the catabolization scheme and the metabolites identified in the algicide, toxic effects may thus be attributed to ammonium, H_2_O_2_ or 4-aminobutanal. The concentration of hydrogen peroxide in the algicide was not significant according to experiments conducted by Pokrzywinski [23], and may not be further considered as the potential toxic metabolite. The toxicity of ammonium to phytoplanktonic cells has been reviewed by Collos and Harrison [35], and highlight the dinoflagellates as the most sensitive class of unicellular algae, with EC_50_ values ranging from 30 to 2700 µM. This toxicity of ammonium has been attributed to the resulting ammonium uptake, suspected to cause intracellular pH disturbance [36] and direct PS II photodamage due to ammonia binding to the Mn complex [37]. Preliminary results to this study indicate that not all dinoflagellate species are subject to impairment of their PS II when exposed to 100 µM of ammonium; only *G. instriatum* presented a significant decrease in its relative fluorescence. These results do not support ammonium as the single toxic agent of the algicide, suggesting that other compound(s) may be involved. The presence of 4-aminobutanal (Figure 2, (**5**)) in IRI-160AA was investigated and barely detectable amounts were detected (Figure 6), revealing a potential implication of this aldehyde in the algicide toxicity. Indeed, naturally occurring aldehydes can be highly harmful to microalgae as demonstrated for the poly unsaturated aldehydes produced by several diatom species [6]. Unfortunately, the instability of the 4-aminobutanal precludes its use as standard in a bioassay, and its toxicity on the dinoflagellates growth was not confirmed. It is worth noting that this instability may yield a spontaneous reduction of the 4-aminobutanal into pyrroline (Figure 2, (**6**)) and H_2_O; however, the presence of pyrroline in the algicide was not determined because of the absence of reaction between dansyl chloride and tertiary amines. On the other hand, concentration of *n*-butylamine in IRI-160AA exceeded naturally occurring levels in surface waters by a factor of 60 [38], suggesting that it originates from *Shewanella* sp. IRI-160 metabolism and plays a key role in the algicidal properties of IRI-160AA. Indeed, *n*-butylamine was shown to inhibit putrescine methyltransferase activity in plants [39], and to be a reaction by-product from 4-aminobutanoate and butanaldehyde in *Pseudomonas* sp. [40]. Although its toxicity on phytoplanktonic species has never been reported, it can be produced as a group of autotoxic compounds by plants [41]. The biochemical mechanisms leading to this important exudation remain, however, undetermined. It is also worth noting here that during the procedure, the algicide is subjected to high temperature and pressure while being autoclaved, potentially altering the chemical structure of metabolites. It thus seems reasonable to suggest that *n*-butylamine could be the by-product of a compound excreted by *Shewanella* sp. IRI-160 in direct link with putrescine catabolization.

Therefore, at this stage of our investigations, only ammonium, 4-aminobutanal, pyrroline or *n*-butylamine, or a synergistic effect of some or all of them may explain the toxicity of IRI-160AA to dinoflagellates. The synergistic effect of ammonium (100 µM) and *n*-butylamine (200 µM), both present in the algicide, was shown to be a good candidate since it resulted in a decrease by 1.37 and 1.47 of the relative fluorescence of the dinoflagellates *P. minimum* and *G. instriatum*, whereas it did not significantly affect the control species *Rhodomonas* sp., in accordance with the mode of action of the algicide. However, to confirm thoroughly the identity of the algicidal compound, further work should include a detailed comparison of the impacts induced by the algicide vs. the synergistic action of ammonia and *n*-butylamine on the cellular morphology and physiology of the tested dinoflagellates [23,24].

The absence of clear identification of the bioactive substance(s) acting in the IRI-160AA highlights the difficulty of characterizing SPC in seawater matrices due to their physico-chemical features, which preclude a facilitated extraction. Yet, such SPC compounds may coordinate basic ecological key-interactions in the marine microbial ecosystem (e.g., diproline [5]), and their characterization would increase our knowledge and understanding of marine microbial interactions. Nowadays, several tools have been made available for the extraction of low molecular weight organic compounds from biological samples [42]: capillary electrophoresis (CE), gas chromatography (GC), and liquid chromatography (LC). To provide the highest level of sensitivity toward metabolite identification and quantification, these separation techniques must be coupled to an on-line spectral detector (i.e., Mass Spectrometry (MS)), allowing metabolomics analysis where relevant. CE is based on the differential transportation of charged species in an electric field through a conductive medium. A wide range of metabolites, from small inorganic ions [43] to large proteins [44], can be separated by CE by adjusting fractionation conditions, like the capillary length, buffer ionic strength, pH, and viscosity. One major disadvantage of CE comes from its unsuitability with MS detection in that it can form micelles that tend to contaminate the ion source, suppress analyte ionization, and decrease MS response. Therefore, only a limited number of methodologies for online CE-MS implementation in biological analysis have so far been developed [42], making it unsuitable for the search of SPC metabolites in seawater. Likewise, classical procedures using GC are time-consuming, due to the prior derivatization steps needed to increase the volatility and stability of SPC and longer time of analysis. Reverse-phase liquid chromatography (RPLC) offers an alternative, although retention and/or selectivity problems may arise for SPC metabolites. Various types of RPLC are currently available, including ion pair chromatography, which offers a robust method for separating SPC compounds such as carbohydrates [45] and polyamines [46]. However, the salts present in seawater may interfere with the retention of the metabolites of interest and the addition of buffers during the procedure of extraction affects MS-based detection by clogging the ion source and reducing the ionization yield of the target analyte. The separation of polar compounds using hydrophilic interaction chromatography (HILIC) has become a common approach over the last decade [47,48]. In HILIC, retention increases with increasing polarity of the stationary phase and solutes, and with decreasing polarity of the organic solvent systems used for elution. For instance, HILIC coupled to MS was recently used to target low molecular weight zwitterionic metabolites in phytoplankton cells [49]. However, even HILIC is not always suitable for SPC metabolites (this study, data not shown). Innovative technologies (e.g., extracting phases) or innovative analytical approaches based on currently available technologies (as for DMS [49]) must be developed to overcome the difficulties associated with the extraction, separation and detection of SPC metabolites in seawater.

## 5. Methods

### 5.1. Algicide Preparation

The preparation of the bacterial filtrate was thoroughly described in [16,25]. Briefly, *Shewanella* sp. IRI-160 was aseptically spread on agar plates and incubated 48 h at room temperature. A bacterial colony was further transferred to a volume of 500 mL of liquid LM medium and incubated at 24 °C on an orbital shaker until reaching an OD of ~2 at 600 nm (NanoDrop 2000 Spectrophotometer; ThermoFisher Scientific, Waltham, MA, USA). The bacterial culture was centrifuged at 5000× *g* for 10 min and resuspended in the same volume of salinity 20, f/2 medium [50]. This wash was repeated and the culture was re-combined in salinity 20, f/2 medium and incubated for 7 days at 24 °C with daily resuspension. After sampling of one milliliter for cell counting, the cultures were subsequently centrifuged at 5000× *g* for 10 min. The supernatant was harvested separately from the cells, autoclaved at 121 °C for 20 min, and further filter-sterilized through 0.2 µm polycarbonate filters and stored at 4 °C until bioassays, preliminary experiments and extraction. To investigate metabolites within the cells, the bacterial pellet was immediately submitted to extraction.

To count the bacterial cells, an aliquot of 10 µL was added to 990 µL acetate buffer (pH 4). Formaldehyde (37%) was added at 2% (*v*/*v*) and the sample was incubated 15–30 min in the dark at room temperature. The sample was further stained using 100 µl of DAPI (4′,6-Diamidino-2-Phenylindole, Dilactate; ThermoFisher Scientific, Waltham, MA, USA; 0.1 mg/mL in phosphate-buffered saline, pH 7.4) and again incubated 15–30 min in the dark at room temperature. Stained samples were filtered onto a 0.2 µm-pore size black polycarbonate filter at very low vacuum and filters were placed on glass slides with mountant solution (Electron Microscopy Sciences, Hatfield, PA, USA) for further microscope analysis. All counting was performed using an EVOS^®^ FL Auto Imaging System (ThermoFisher Scientific). Cell density was calculated as in [51].

### 5.2. Algal Cultures and Bioassays

Stock cultures of *Gyrodinium instriatum* (CCMP 2935, [National Center for Marine Algae and Microbiota, https://ncma.bigelow.org/]), *Prorocentrum minimum* (CCMP 2233), and *Rhodomonas* sp. (CCMP 757; cryptophyte) were maintained in natural sea water with f/2 nutrients (−Si) [50] and a salinity of 20, under 25 °C, and with a light intensity of approximately 130 µmol photons m^−2^·s^−1^. The cultures were kept under a 12 h:12 h light: dark cycle, and semi-continuously in exponential growth phase.

Bioassays were conducted by incubating an aliquot of the microalgal culture in culture tubes at a final concentration of 50%, by adding fresh f/2 medium, salinity 20. The addition of the algicide at 10% of the final concentration was further performed at the beginning of the exponential phase (usually after 3–4 days following [16] protocol) and the algal photochemistry was assessed after 24 h by measuring the in vivo active chlorophyll fluorescence (FASTtrack II). Indeed, the algal fluorescence was previously shown to be a good proxy of the algal abundance [16]. The conditions of the measurement are detailed in [25].

### 5.3. Preliminary Experiments

Several aliquots of the fresh sterile algicide were sampled to undertake specific preliminary experiments:

Size: the algicide size was assessed using Float-A-Lyzer G2 Dialysis Device (Spectrum Laboratories, Rancho Dominguez, CA, USA) having a 100–500 Da cut-off and previously washed in 10% ethanol for 10 min and MQ water for 20 min following the manufacturer’s recommendations. Two replicates of 10 mL of the algicide were poured in two 10 mL-dialysis tubes already placed in large beakers filled with MQ water on magnetic stir plates. The diffusion of the algicide was carried out over 21 h while being stirred at low speed. The contents of the dialysis tubes were subsequently retrieved and stored at 4 °C for use in bioassays.

Stability: the resistance to heat had previously been assessed by Pokrzywinski [16] using capped containers. Yet, we noticed that heating the algicide until close to dryness or full dryness significantly decreased its activity (*p* < 0.01). Therefore, several approaches to evaporation were evaluated: 30 mL of algicide was evaporated to dryness using (1) a rotavap at 35–40 °C, (2) an autoclave at 121 °C for 20 min, (3) a hot plate at 90 °C with a hot-water bath. The dried algicide was subsequently re-suspended in 1 mL of either DMSO or MQ water, concentrating the algicide by a factor 30. Seawater was avoided for resuspension due to high concentration of salts after evaporation of the algicide. Similar evaporation was conducted using f/2 medium as a control. The activity of the algicide and control resuspensions was further tested using the bioassay described above.

### 5.4. Extraction and Chromatographic Analysis

Extraction of amines in both the bacterial cells and the algicide: The bacterial culture was centrifuged at 5000 × *g* for 10 min, and the cells and the supernatant were harvested and extracted separately. The cell pellet was subsequently rinsed by resuspension in salinity 20 f/2 medium and further centrifuged at 5000 × *g* for 10 min. The pellet was extracted for 15 min in ultrasonic bath with 1 mL of 6N HCL and the extract stored at 4 °C until further derivation.

The culture supernatant was filtered through a 0.2 µm polycarbonate filter and stored at 4 °C until further treatment. After unsuccessful results in our attempt to isolate the algicidal compound(s) using several types of SPE cartridges (C18 silica Sigma Aldrich, St Louis, MO, HLB^®^ Waters, Milford, MA, HRX^®^ Macherey Nagel, Bethlehem, PA, USA) under various conditions (acidic, neutral, basic pH) and solvents for the elution (methanol, acetonitrile, mixture of water/methanol/acetonitrile), we used alternative protocols. Since the data (volatility and odor) suggested that the algicidal exudate likely included compounds in the amines family, dansyl chloride was used as a derivative agent. Dansyl chloride (Sigma Aldrich, St Louis, MO, USA) was first solubilized in acetone at a concentration of 5 g/L and 500 µL were added in a 20 mL glass vial to 1 mL of the algicide extract previously brought to pH 10.5 using 1 mL of NaHCO_3_. A similar procedure was followed for the cell pellet extract except 3 mL of NaHCO_3_ as well as a substantial volume (~1.5 mL) of 5M NaOH were added to reach pH 10. The vial was covered with aluminum foil, slightly unscrewed, and immediately heated at 100 °C for 12 min. Subsequently, 500 µL of glycine (100 mg/mL) was added to the mixture and further incubated in the oven at 100 °C for 6 min to react with the excess of dansyl chloride and therefore stop the reaction. The vial was transferred to a fume hood and uncapped to allow evaporation of the acetone for 2 h. The remaining aqueous phase was extracted at very low vacuum (mostly by gravity) onto a C18 silica cartridge (200 mg, Sigma Aldrich, St Louis, MO, USA) observing the following steps: (i) activation with 1 mL of acetonitrile, (ii) equilibration with 1 mL of MQ water pH 11, (iii) sample load, (iv) rinsing with 1 mL of MQ water pH 11 and (v) elution with 1.5 mL of acetonitrile. The eluate was filtered through a 0.2 µm syringe filter and evaporated to a volume of 500 µl before being stored at −20 °C until HPLC-fluorescence and UHPLC-HRMS analysis. Standards of *n*-butylamine and putrescine were also purchased (Sigma Aldrich, St Louis, MO, USA) and derivatized following the same procedure to allow identification by HPLC-fluorescence and UHPLC-UV-HRMS, respectively.

Presence of aldehydes in the algicide and the cells: the presence of aldehydes in both the cell pellet and the supernatant was investigated using 2,4 DNPH as a derivative agent. Crystals of 2,4-DNPH (Sigma Aldrich, St Louis, MO, USA) were obtained by boiling twice 3 g of commercial 2,4 DNPH in a water bath placed on a hot plate and further rinsed twice using acetonitrile (ACN) before being recrystallized. An amount of 20 mg of recrystallized DNPH was solubilized into 15 mL of HCL:ACN:H_2_O (2:1:5). The solution was subsequently extracted with hexane to remove impurities from the mixture and the organic layer was discarded. The aqueous layer was used as the derivative solution: 500 µL was added to 1 mL of the (i) cellular extract, (ii) algicide and (iii) a standard of butanal (butyraldehyde), before visualization of putative precipitation of aldehydes.

Chromatographic analysis: Chromatographic analysis was first performed on a Shimadzu system (PU-LC10ADVP equipped with a Shimadzu RF-10AXL Fluorescence detector; Shimadzu Scientific Instruments, Columbia, MD, USA). Fluorescence was set at 330 nm for excitation and 530 nm for emission wave lengths. Chromatographic separation was achieved on a C18 column (XTerra 5 µm, 4.6 × 50 mm, Waters) using a linear elution gradient of H_2_O/CH_3_CN/formic acid from 70:30:0.01 (*v*/*v*/*v*, isocratic from 0 to 20 min) to 0:100:0.1 (*v*/*v*/*v*, isocratic from 20 to 30 min) at a flow rate of 1 mL/min. The injected volume was 20 µL. Further on-line UHPLC-UV-HRMS analysis were performed on a Q-Exactive Orbitrap mass spectrometer interfaced with a Dionex ultimate 3000 UHPLC system (Thermo Scientific, Waltham, MA, USA). Separation of the metabolites was achieved by reversed-phase high performance liquid chromatography (RP-HPLC) (Waters, ACQUITY UPLC BEH C18 1.7 µm, 50 mm × 2.1mm) using a stepwise elution of H_2_O/CH_3_CN/formic acid from 85:15:0.1 (*v*/*v*/*v*) to 0:100:0.1 in 8 min, at a flow rate of 0.5 mL/min. The first minute of the LC run was diverted to waste to prevent salts from reaching the mass spectrometer. MS data were acquired in the positive mode at the 300 to 2000 *m*/*z* range, using a data-dependent top 5 method dynamically choosing the most abundant precursor ions from the survey scan for HCD fragmentation using a stepped normalized collision energy of 35, 40, 45 eV. Survey scans were acquired at a resolution of 70,000 at *m*/*z* 200 on the Q Exactive. The spectrometer analyzer parameters were set as follows: dry gas, N_2_ (2 L/min); capillary temperature, 300 °C; capillary voltage, 3.500 kV; end plate offset, 500 V; collision gas, He; collision energy, 7 eV.

Several standards were run together with the samples in order to ensure thorough characterization of the algicidal compounds: 4-aminobutanal (1.8 M), dansylated putrescine, spermidine, ammonium chloride and *n*-butylamine at a concentration of 1 mg/mL, each. The standard of 4-aminobutanal was obtained from 100 µl of the 4-aminobutanal diethyl acetal solution (Sigma Aldrich, St Louis, MO, USA) treated with 2 mL of MQ water and subsequently concentrated at 37 °C to a final volume of 800 µL. An aliquot of the concentrate (20 µl) was dissolved in 5.5 mL of 0.07N HCL to ensure the hydrolysis of the acetal, giving a 1.8 M solution of 4-aminobutanal.

### 5.5. Validation

Effects of *n*-butylamine and ammonium on dinoflagellates and cryptophytes: To investigate the effects of *n*-butylamine, ammonium, and their combination on dinoflagellates and *Rhodomonas* sp. (cryptophyte; control species), stock cultures in exponential growth phase were divided into 30 mL glass culture tubes (N = 3 in each treatment). One hundred micromolar of NH_4_Cl was added to one treatment, either 20 µM or 200 µM *n*-butylamine (Thermo Fisher Scientific, Waltham, MA, USA) was added to one treatment, and a combination of 100 µM NH_4_Cl and 20 µM or 200 µM *n*-butylamine was added to one treatment, and same amount of sterile MilliQ water was added to controls. The treatments and controls were then incubated under the same condition as the stock cultures for 24 h, and in vivo fluorescence of chlorophyll *a* was measured as a proxy for algal cell density [24]. To assess the difference in cell density between treatments and controls, relative fluorescence was calculated as a ratio of in vivo fluorescence of chlorophyll *a* in treatments to the average in vivo fluorescence of controls.

Statistical analyses: The significant difference in relative fluorescence between treatments was tested by one-way ANOVA. If the effects of the same chemical or combination of chemicals across species were significantly different (*p* < 0.05), then TukeyHSD test was conducted to test the significance of difference in relative fluorescence between all possible groups.

## Figures and Tables

**Figure 1 molecules-24-00135-f001:**
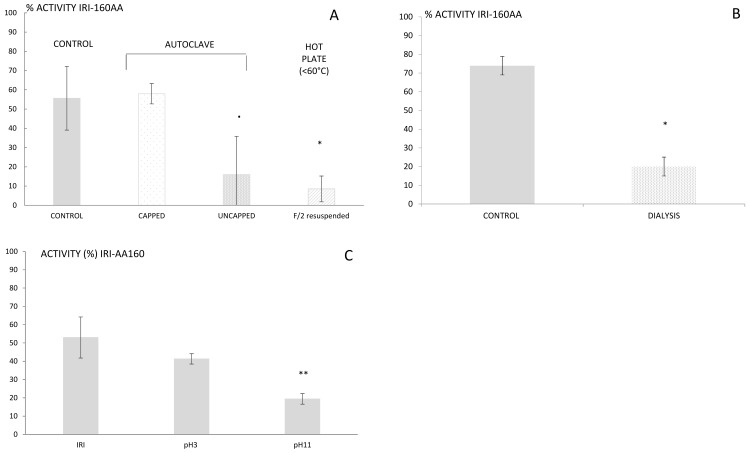
Activity of the algicide IRI-160 in % after being submitted to various treatments: heat (**A**), dialysis bags (**B**), and pH modification (**C**). All bioassays were done using *G. instriatum* and *Rhodomonas* sp. as positive and negative control species, respectively. Statistical significance is indicated by **, *, and · for *p* < 0.005, <0.05 and <0.1, respectively.

**Figure 2 molecules-24-00135-f002:**
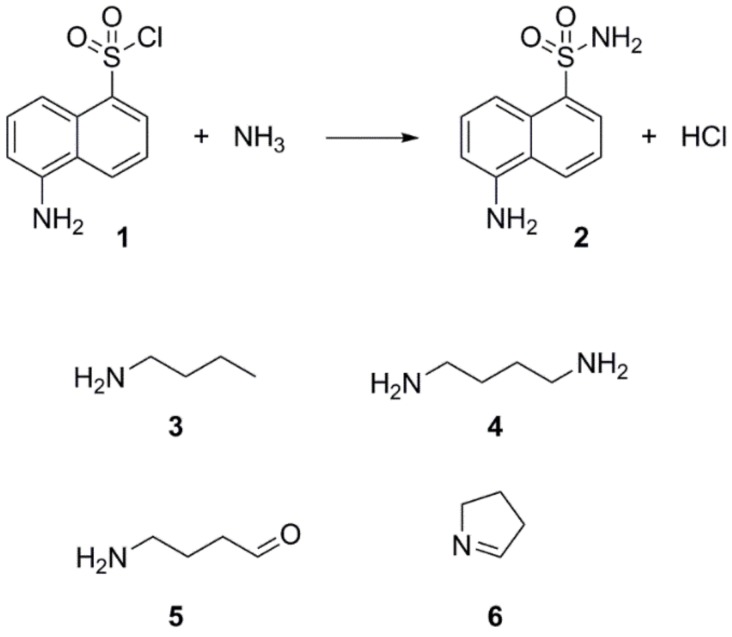
Reaction scheme involving dansyl chloride (**1**) and dansylated ammonium (**2**), and the chemical structures of the amino compounds identified in this study (**3** to **6**).

**Figure 3 molecules-24-00135-f003:**
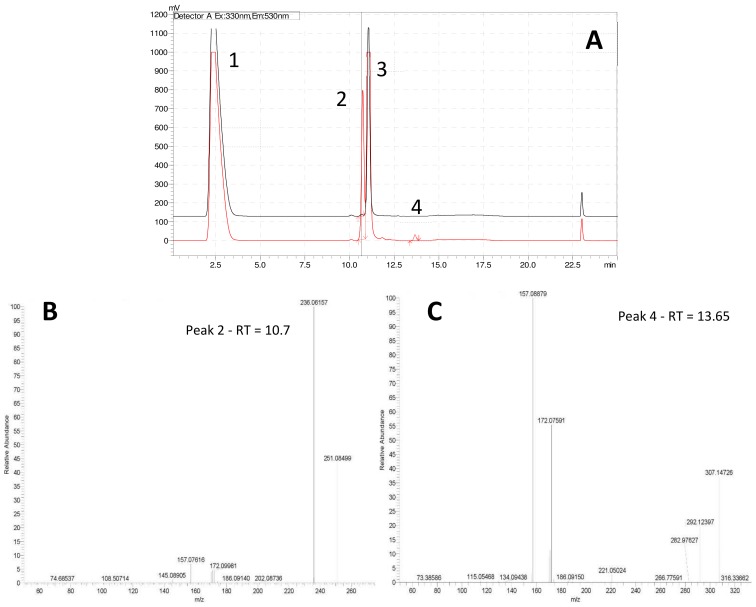
(**A**) HPLC-fluorescence chromatogram comparing f/2 blank (black line) and IRI-160AA (red line): 1 = DNS-OH, 2 = DNS-NH_3_, 3 = DNS-glycine, and 4 =DNS-*n*-butylamine. Lower panels: Fragmentation patterns (MSMS spectra) of peaks 2 and 4 confirming the presence of DNS-NH_3_ (**B**) and DNS-*n*-butylamine (**C**).

**Figure 4 molecules-24-00135-f004:**
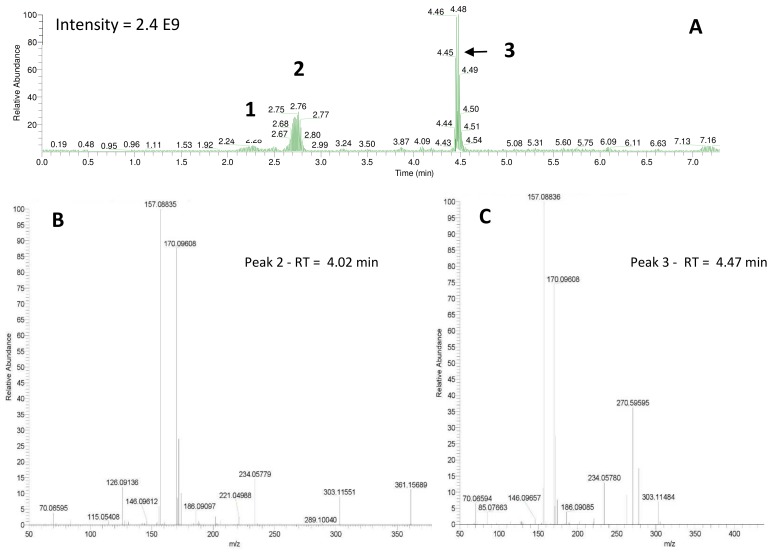
(**A**) UHPLC-HRMS chromatogram of the *Shewanella* 160 cells previously derivatized: 1 = DNS-NH_3_, 2 = DNS-glycine, and 3 = DNS-putrescine. Lower panels: Fragmentation patterns (MSMS spectra) of peaks 2 and 3 confirming the presence of DNS-4-aminobutanal (**B**) and DNS-putrescine (**C**).

**Figure 5 molecules-24-00135-f005:**
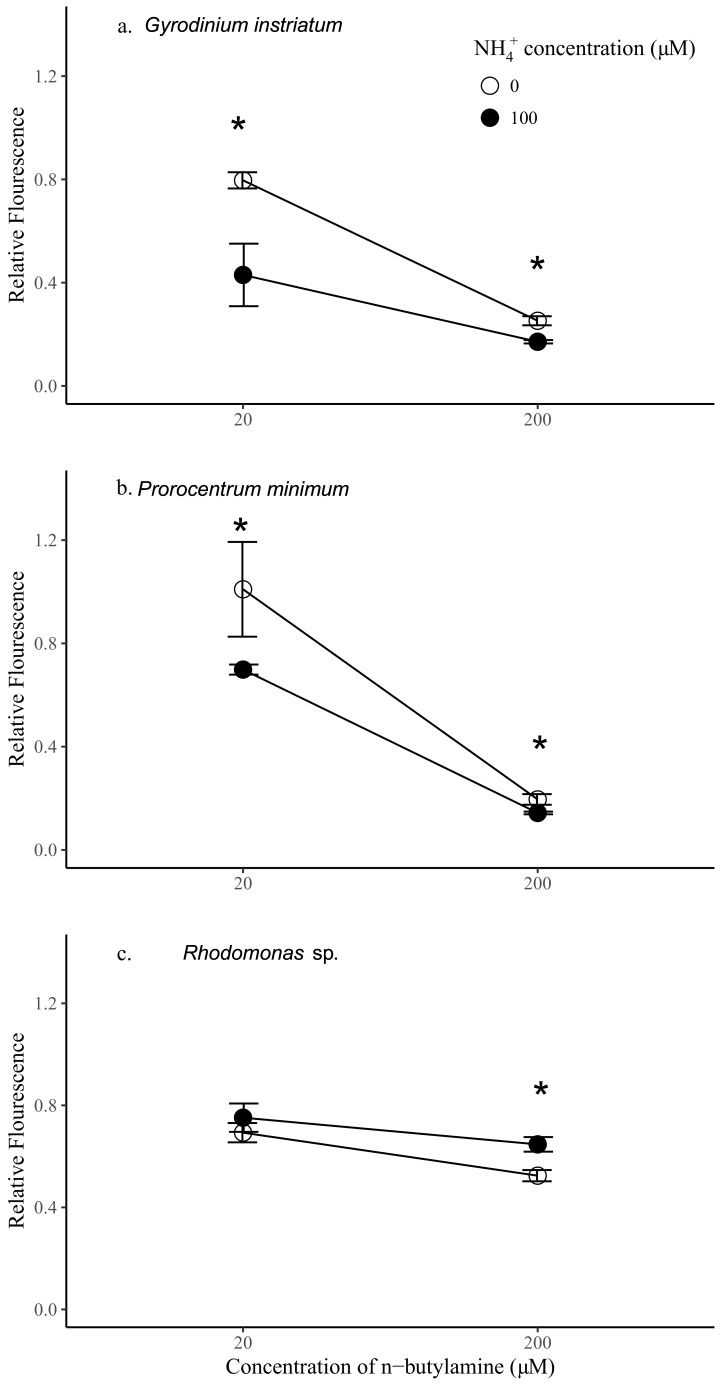
Relative fluorescence of *Gyrodinium instriatum* (**a**), *Prorocentrum minimum* (**b**), and *Rhodomonas* sp. (**c**) after 24 h exposure to 20 µM *n*-butylamine, 200 µM *n*-butylamine, 20 µM *n*-butylamine + 100 µM NH_4_^+^, and 200 µM *n*-butylamine + 100 µM NH_4_^+^. Asterisks indicate significant differences in relative fluorescence between *n*-butylamine alone and *n*-butylamine + NH_4_^+^ treatments.

**Figure 6 molecules-24-00135-f006:**
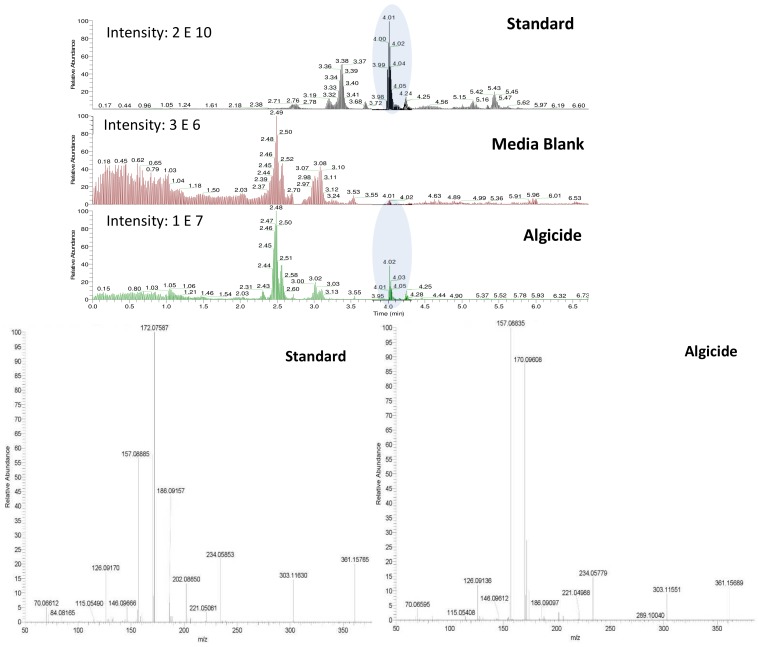
UHPLC-HRMS chromatograms of a standard of 4-aminobutanal at a concentration of 180 µM, as well as extracted chromatogram showing ions in the range *m*/*z* 361.15-361.16 in f/2 media blank and the algicide. A comparison between the fragmentation patterns (MSMS spectra) of 4-aminobutanal retrieved in the standard and in the algicide is also given for confirmation.

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
