# Peer review of "Small Polar Molecules: A Challenge in Marine Chemical Ecology"

_molecules, 2018, doi:10.3390/molecules24010135_

Round 1
Reviewer 1 Report
"Very small molecules: a challenge in marine chemical ecology" attempts to describe the isolation of algicidal metabolites from Shewanella sp. IRI-160. The premise of the manuscript is that there are two challenges in the isolation and characterization of bioactive metabolites - small size and polarity. While this reviewer would agree with the later there are ample examples in the literature of small bioactive metabolites isolated from many different organisms. Polar molecules have been a challenge, however there are many studies now that target these molecules. This reviewer recognizes that a considerable amount of bench work has been conducted and will be more valuable to the reader if it is presented better.
Pg. 2, lines 51-68The introduction is too long. There are many studies that specifically report the isolation and characterization of small water soluble molecules with ecological activities. The discussion of bioassay-guided fractionation is disjointed and hard to follow. We are a long way away from just adopting techniques from drug discovery in the marine chemical ecology fields. Please look at the literature regarding the study of chemical cues in algae, corals and other invertebrates. There was also a review article by Joe Pawlik and colleagues that clearly looks at the evolution of techniques in this field.
Pg 2., Lines 69-79 The yield of metabolites is a real issue and will be a better talking point than the size of the molecules. Most of this paragraph is a jumble of ideas without cohesiveness.
Section 2, analytics procedure. This section reads like a recipe without any clear ingredients or procedures to prepare something. It is a bunch of "what ifs" that appear to justify the use of the steps. There is ample info in the results that should be brought back to this section that would allow a reader to copy the experiments in their own lab.
The results section is through and should be backed up well with the procedure section. This section is missing STRUCTURES. Also, a discussion needs to be added that talks about the potential to chemically alter the natural products with all the of the procedures that were used in the study.
This reviewer is unable to determine if the conclusions are valid because the methods are not clear in the manuscript. This manuscript requires significant editing before publication.
Author Response
Pg. 2, lines 51-68The introduction is too long. There are many studies that specifically report the isolation and characterization of small water soluble molecules with ecological activities. The discussion of bioassay-guided fractionation is disjointed and hard to follow. We are a long way away from just adopting techniques from drug discovery in the marine chemical ecology fields. Please look at the literature regarding the study of chemical cues in algae, corals and other invertebrates. There was also a review article by Joe Pawlik and colleagues that clearly looks at the evolution of techniques in this field.
The introduction was modified accordingly. Several additions and removal were done: lines 45-65 “Extraction and further characterization of active metabolites from seawater can be extremely challenging considering (i) a highly ionized matrix inherent to the chemistry of seawater, and (ii) the low yield of metabolites isolated that would be available for spectral analysis (e.g. Nuclear Magnetic Resonance).
Metabolite characteristics lead to a wide range of specific behavior in seawater that prevents the establishment of a universal extraction procedure. Although significant advances in the characterization of marine chemical cues from bacteria11, microalgae12, copepods7, algae12 and invertebrates13 have been accomplished the past decades, very few of these metabolites are of very low molecular weight3,11 (C1-C6, <125 Da) and extreme polar features5,14. This extreme polarity contributes to the inability for chemical characterization of active metabolites in a number of cases15–17. Yet, several of these Small Polar Compounds (SPC) secreted by microbial organisms were shown to play key roles in biotic interactions, such as small oxylipins6, amino-acids (e.g. N-β-methylamino-L-alanine (BMMA)18), pheromones (e.g. L-diproline5), polyamines (e.g. spermine and putrescine19), and volatile compounds (e.g. dimethylsulfide20). Clearly, there is a need to resolve the analytical bottlenecks preventing the characterization of SPC (< 125 Da) dissolved in seawater.
In this context, classic methods for extraction (i.e liquid-liquid extraction (LLE) and solid phase extraction (SPE)) of these metabolites have little value, and specific analytical methods must be developed to target the metabolite of interest. Comparative mass spectrometry-based (MS) metabolomics was recently presented as a good substitute for the characterization of metabolites since it does not require the fractionation nor the isolation steps5,21, however, extraction from seawater matrices makes the use of fractionation tools compulsory in this case.“
Pg 2., Lines 69-79 The yield of metabolites is a real issue and will be a better talking point than the size of the molecules. Most of this paragraph is a jumble of ideas without cohesiveness.
This paragraph was removed.
Section 2, analytics procedure. This section reads like a recipe without any clear ingredients or procedures to prepare something. It is a bunch of "what ifs" that appear to justify the use of the steps. There is ample info in the results that should be brought back to this section that would allow a reader to copy the experiments in their own lab.
This part of the ms was removed and now backs up the results section.
The results section is through and should be backed up well with the procedure section. This section is missing STRUCTURES.
A reaction scheme was added to the Results section and structures of all mentioned metabolites were added to the Discussion.
Also, a discussion needs to be added that talks about the potential to chemically alter the natural products with all the of the procedures that were used in the study.
This discussion section was modified to include this discussion: line 356-364 “Indeed, n-butylamine was shown to inhibit putrescine methyltransferase activity in plants39, and to be a reaction by-product from 4-aminobutanoate and butanaldehyde in Pseudomonas sp.40. Although its toxicity on phytoplanktonic species has never been reported, it can be produced as in a group of autotoxic compounds by plants41. The biochemical mechanisms leading to this important exudation remain, however, undetermined. It is also worth noting here that during the procedure, the algicide is subjected to high temperature and pressure while being autoclaved, potentially altering the chemical structure of metabolites. It seems thus reasonable to suggest that n-butylamine could be the by-product of a compound excreted by Shewanella sp. IRI-160 in direct link with putrescine catabolization”.
Reviewer 2 Report
This paper report is about the characterization of bioactive metabolites obtained from bacteria Shewanella sp. The manuscript is written in good style. The manuscript fits into the Journal’s aims and scope, and I think it is interesting enough to be published. However, there is a lot of missing information that needs to be completed. Based on the detailed comments below, I suggest a major revision of the MS.
Specific comments:
Please, unify the record of units throughout the text (i.e. ml or mL)
Please check the literature for errors (i.e., No. 40). Species should be written in italics.
Please improve the quality of all the figures. The scale is too small, I can not see the value. Please prepare the figures in an appropriate way to fit the standards published in good Journals with high IF.
L34 – I would add information about allelopathy.
L47 – production of secondary metabolites may also depend on other factors, i.e. light intensity, salinity, availability of nutrients (e.g., Antunes et al., 2012; Śliwińska-Wilczewska et al., 2016; Śliwińska-Wilczewska and Latała, 2018).
L48 – The problem is also the fact that different organisms (e.g., cyanobacteria and microalgae) can produce a number of different compounds (allelochemicals), and their activity may depend on their concentration and mutual proportion.
L50 – I think it is worth getting acquainted with the work of Leão et al. (2012).
L81 – I think it is worth giving specific names of dinoflagellates species.
I think that the Chapter “Analytical Procedure” should be included in the Introduction section or removed from the MS. The information in this chapter raises many questions. The reader may have the wrong impression that it is part of the Materials and Methods section.
L355 – „To investigate metabolites within the cells, the bacterial biomass was immediately submitted to extraction”. – This is a very important issue. It should be given in the text how many bacteria were at the beginning of the experiment. The authors state in Abstract that Shewanella sp. IRI-160 produce 0.05-6.5 fmol/cell of small amines. How was it counted?
L367 – How did the authors determine based on fluorescence that tested microalgae were in the exponential growth phase?
L459 – Maybe it's better to change the subtitle to: “Effects of n-butylamine and ammonium on dinoflagellates and cryptophytes”?
L220 – I would like to see these results.
L253 – “Extracts of the bacterial cells do not hold similar toxicity (data not shown), precluding the extraction of the bioactive metabolite(s) from the cells and necessitating extraction of bioactive compounds from the culture medium”. Why these results were not shown? Maybe it would be good to add them to the supplement. It seems strange that the extracts show different toxicity. I would expect a reverse dependence. Extracts should show similar toxicity, and the activity of compounds released into the medium should be characterized by lower repeatability.
Author Response
Please, unify the record of units throughout the text (i.e. ml or mL)
All units were unified
Please check the literature for errors (i.e., No. 40). Species should be written in italics.
The literature was checked and corrected accordingly
Please improve the quality of all the figures. The scale is too small, I can not see the value. Please prepare the figures in an appropriate way to fit the standards published in good Journals with high IF.
All figures were modified to improve their quality
L34 – I would add information about allelopathy.
Lines 33-39, the text was modified as following: “Allelopathic interactions are now recognized to contribute to the paradoxical coexistence of a large diversity of species in an ecosystem with limiting resources as evoked by 4 in the “paradox of the plankton”. In the microbial ecosystem, bacteria and microalgae secrete secondary metabolites to mediate various interactions like competition for resources, mate attraction or predation (3 for a review). Microbial allelochemicals …”
L47 – production of secondary metabolites may also depend on other factors, i.e. light intensity, salinity, availability of nutrients (e.g., Antunes et al., 2012; Śliwińska-Wilczewska et al., 2016; Śliwińska-Wilczewska and Latała, 2018).
We totally agree with the reviewer but we believe this is not the scope of our paper.
L48 – The problem is also the fact that different organisms (e.g., cyanobacteria and microalgae) can produce a number of different compounds (allelochemicals), and their activity may depend on their concentration and mutual proportion.
Yes, we totally agree with the reviewer and this is the reason why mixtures of compounds were tested as shown in results and in the discussion
L50 – I think it is worth getting acquainted with the work of Leão et al. (2012).
We thank the reviewer for bringing this work to our attention. The citation was added.
L81 – I think it is worth giving specific names of dinoflagellates species.
All names of dinoflagellates were added
I think that the Chapter “Analytical Procedure” should be included in the Introduction section or removed from the MS. The information in this chapter raises many questions. The reader may have the wrong impression that it is part of the Materials and Methods section.
We removed the whole “Analytical Procedure” chapter and included it to the results part.
L355 – „To investigate metabolites within the cells, the bacterial biomass was immediately submitted to extraction”. – This is a very important issue. It should be given in the text how many bacteria were at the beginning of the experiment. The authors state in Abstract that Shewanella sp. IRI-160 produce 0.05-6.5 fmol/cell of small amines. How was it counted?
We thank the reviewer for pointing this oversight. The procedure for bacteria cells counting was added to Material and Methods: “To count the bacterial cells, an aliquot of 10 µl was added to 990 µl acetate buffer (pH 4). Formaldehyde (37%) was added at 2% (v/v) and the sample was incubated 15-30 min in the dark at room temperature. The sample was further stained using 100 µl of DAPI (4',6-Diamidino-2-Phenylindole, Dilactate; ThermoFisher Scientific, MA, US; 0.1 mg ml-1 in phosphate buffered saline, pH 7.4) and again incubated 15-30 min in the dark at room temperature. Stained samples were filtered onto a 0.2 µm-pore size black polycarbonate filter at very low vacuum and filters were placed on glass slides with mountant solution (Electron Microscopy Sciences, PA, US) for further microscope analysis. All counting was performed using an EVOS® FL Auto Imaging System (ThermoFisher Scientific). Cell density was calculated as in51.”
Besides, the optical density (OD) measured for the bacterial culture is given in the text line 429.
L367 – How did the authors determine based on fluorescence that tested microalgae were in the exponential growth phase?
This was showed by Pokrzywinski et al. in 2012 and we based our experiments on their findings. The reference was added to the text for clarification.
L459 – Maybe it's better to change the subtitle to: “Effects of n-butylamine and ammonium on dinoflagellates and cryptophytes”?
The title was changed following the reviewer’s recommendations
L220 – I would like to see these results.
A supplemental file was added to the paper to show the preliminary results obtained for the addition of ammonia + n-butylamine on all species tested.
L253 – “Extracts of the bacterial cells do not hold similar toxicity (data not shown), precluding the extraction of the bioactive metabolite(s) from the cells and necessitating extraction of bioactive compounds from the culture medium”. Why these results were not shown? Maybe it would be good to add them to the supplement. It seems strange that the extracts show different toxicity. I would expect a reverse dependence. Extracts should show similar toxicity, and the activity of compounds released into the medium should be characterized by lower repeatability.
One reason for the difference of toxicity between the cells extract and the algicide could be linked to either the absence of a de novo production or an alteration of the chemical structure of the metabolites excreted by the bacteria in the algicide (during autoclaving for instance). A sentence was added lines 342-346: “It worth be noted here that during the procedure, the algicide is submitted to high temperature and pressure while being autoclaved, potentially altering the chemical structure of metabolites. It seems thus reasonable to suggest that n-butylamine could be the by-product of a compound excreted by Sheawanella sp. IRI-160 in direct link with putrescine catabolization cycle.” On the other hand, lines 298-300 and 306-308 were removed as we realized they could be confusing for the reader.
Round 2
Reviewer 1 Report
The authors adequately responded to the comments. The methods in the study may be useful to other researchers.
Reviewer 2 Report
I believe the manuscript has been significantly improved and now warrants publication in Molecules.